# Generation of scalable cancer models by combining AAV-intron-trap, CRISPR/Cas9, and inducible Cre-recombinase

Prajwal C. Boddu[1,5], Abhishek K. Gupta[1,5], Jung-Sik Kim[2], Karla M. Neugebauer[3], Todd Waldman[2] & Manoj M. Pillai[1,4 ✉]

Scalable isogenic models of cancer-associated mutations are critical to studying dysregulated gene function. Nonsynonymous mutations of splicing factors, which typically affect one allele, are common in many cancers, but paradoxically confer growth disadvantage to cell lines, making their generation and expansion challenging. Here, we combine AAV-intron trap, CRISPR/Cas9, and inducible Cre-recombinase systems to achieve >90% efficiency to introduce the oncogenic K700E mutation in SF3B1, a splicing factor commonly mutated in multiple cancers. The intron-trap design of AAV vector limits editing to one allele. CRISPR/Cas9-induced double stranded DNA breaks direct homologous recombination to the desired genomic locus. Inducible Cre-recombinase allows for the expansion of cells prior to loxp excision and expression of the mutant allele. Importantly, AAV or CRISPR/Cas9 alone results in much lower editing efficiency and the edited cells do not expand due to toxicity of SF3B1-K700E. Our approach can be readily adapted to generate scalable isogenic systems where mutant oncogenes confer a growth disadvantage.

[1] Section of Hematology, Yale Cancer Center, Yale University School of Medicine, New Haven, CT, USA. [2] Department of Oncology, Molecular Biology and Genetics, Lombardi Cancer Center, Georgetown University, Washington, DC, USA. [3] Department of Molecular Biophysics and Biochemistry, Yale University School of Medicine, New Haven, CT, USA. [4] Department of Pathology, Yale University School of Medicine, New Haven, CT, USA. [5]These authors contributed equally: Prajwal C. Boddu, Abhishek K. Gupta. ✉email: manoj.pillai@yale.edu

sogenic cell line systems of non-synonymous mutations are essential to understanding the biochemistry and cellular consequences of disease mutations. The CRISPR/Cas9 system has revolutionized genomic editing for purposes of generating both in vitro and in vivo model systems[1–3], but challenges of its adaptation to some scenarios remain. Such scenarios include generation of single-allele missense point mutations or when the resultant mutation confers a physiological growth disadvantage. Targeted double-stranded breaks (DSBs) created by clustered regularly interspaced short palindromic repeats (CRISPR)/ CRISPR-associated protein 9 (Cas9) system are typically repaired by non-homologous end joining[4,5] introducing gene knock-outs. By contrast, gene knock-ins of point mutations require a donor DNA template for homology-directed repair (HDR), a very low efficiency process often corrupted by additional target-site indels[6], necessitating screening of multiple clones to isolate those with correct genome editing[7,8]. Additional difficulties arise when the introduced mutation is toxic to the cell resulting in loss of cell viability and/or loss of mutant allele on prolonged culture[9], which significantly limits scalability of these edited cell systems. To exemplify this, we had tried to generate a *SF3B1*K700E mutation in K562 cell lines using a conventional CRISPR/Cas9 with linear ultramer template but were unable to successfully isolate edited clones despite screening several hundreds of them. It is likely that this was a consequence from the toxicity of the mutated protein.

Many disease-associated mutations, such as those involving splicing factors[10], are heterozygous and paradoxically result in slower growth or cellular death in in vitro model systems while promoting clonal evolution in vivo[9,11,12]. This significantly limits our ability to keep them viable in culture and to investigate molecular mechanisms of disease, particularly in scenarios where a large number of cells are required. There is therefore an unmet need in the field to generate highly efficient in vitro isogenic model systems in which expression of the toxic mutation can be regulated temporally.

Adeno-associated virus (AAV)-based approaches have long been used for creation of gene sequence-specific knock-ins into endogenous alleles in tissue culture-based systems[13]. Recombinant AAV (rAAV) are replication-deficient genetic-payload vehicles generated by replacing the entire AAV genome except for the inverted terminal repeats (ITRs), which are essential for viral genome encapsidation and in vitro viral production[14]. Following cellular attachment and entry of encapsidated rAAV into cells, they exist mainly as extra-chromosomal episomes and rarely integrating into the genome[15]. Since the initial report of rAAV vectors as a promising gene-editing system in 1998[16], several groups have worked to improve the efficiency of this gene targeting approach by introducing gene-trap systems into targeting constructs[17–19]. Using a neomycin resistance (Neo^R; G418)-based intron-trap gene-editing system, the Waldman laboratory has previously reported on the use of rAAV gene editing to introduce epitope tags[18] as well as oncogenic point mutations[20] into endogenous alleles of genes in human cell lines. However, the targeting efficiency of gene editing with this approach is typically modest, in the range of 1–40% among G418-selected clones[18,20]. The antibiotic selection of these resistant clones may be attributed to low frequency, random integration of the rAAV genomes (and not necessarily targeted to desired genomic coordinates), despite consisting of one Kilobyte (Kbp) long homology arms[21].

Previous studies have combined rAAV- and targeted nuclease-based (such as CRISPR/Cas9) approaches to improve upon the efficiency of either approach when deployed alone[22–24]. In these studies, the rAAV vector provides a long single-stranded (ss) DNA template for HDR following double-stranded DNA breaks introduced by CRISPR/Cas9. Initial efforts that combined

CRISPR/Cas9 components and HDR template in the same vector were limited by the maximum length that rAAV vectors could accommodate (~4.5 Kb between ITR). Bak et al.[25] first demonstrated the feasibility of delivering CRISPR components as ribonucleoproteins (RNPs) thereby freeing up the space in the rAAV to fill with the repair template. Subsequently, it was reported that combining RNP and ss linear rAAV donor delivery increased the efficiency of gene editing by up to tenfold over RNP and double-stranded conventional plasmid donor systems[26]. Additional adaptations of the rAAV-CRISPR gene-editing methodology with varied in vivo and in vitro applications have since been reported[24,25,27–30]. Strategies requiring targeted transgene insertions specifically at the DNA break site may however be undesirable for some applications requiring scarless gene editing; integrating AAV targeting constructs typically contain a selection marker, which leaves a scar upon loxp excision in the final cell lines[13].

Targeting gene alterations that do not confer a selectable growth advantage and subsequent recovery of knock-in cells requires a high-efficiency gene-editing strategy. In this regard, advances in rAAV vector gene targeting, through the creation of promoter trap targeting vectors carrying gene entrapment cassettes termed synthetic exon promoter trap (SEPT)[31], have greatly simplified and improved upon the rAAV gene-editing efficacy. The pAAV-SEPT-Acceptor plasmid-based (Addgene # 25648) HDR template used in our model contains a SEPT cassette, which begins with a short intron sequence followed by an IgG variable region-derived splice acceptor, a IRES that permits the translation of the Neo^R open reading frame from RNA transcripts initiated upstream, and a polyadenylation site[32,33]. The intron-trap cassette is flanked by tandem LoxP sites oriented so that expression of the Cre-recombinase results in excision of all sequences within them. Despite such high efficiency of rAAV-HDR, a significant problem with the approach is that of non-specific integration of the repair template in the transcribed regions of the genome that will generate Neo-resistant clones that do not contain the desired modification. We postulated that if the HDR template is directed to the correct genomic locus, the proportion of correct clones can be vastly improved. To this purpose, we combined the rAAV approach with CRISPR/Cas9 to facilitate high-precision genome editing. This approach is an extension of our previously developed inducible AAV-based intron-trap system. We demonstrate that combining rAAV and CRISPR/Cas9 is feasible and leads to much higher efficiency than attainable by rAAV alone. Additionally, we use inducible expression of Cre-recombinase to temporally control expression of mutant allele, thereby, overcoming problems of toxicity of the mutant protein.

## Results

**Genome editing with rAAV alone for single base pair changes results in low efficiency editing without incorporating the desired mutation.** The lysine to glutamic acid substitution at position 700 (K700E) is the most common non-synonymous mutation described in multiple clonal processes such as myelodysplastic syndromes, chronic lymphocytic leukemia, uveal melanoma, lung cancer and breast cancer[34]. Like other splicing factor mutations, *SF3B1* mutations are heterozygous and non-synonymous suggesting a gain-of-function change. It is therefore critical to study these mutations in isogenic model systems, and not in overexpression models where the proportion of wild-type to mutant protein is unlikely to be equal.

The rAAV model with intron trap, as we have previously described, is ideal for this purpose given that it can specifically select clones edited at a single allele. To introduce the K700E mutation in exon 15 of human *SF3B1* gene locus, we sequentially

cloned the two homology arms into the polylinkers that had been built into the pAAV-SEPT-Acceptor plasmid (Fig. 1: steps 1 and 2). The additional screening primer was also introduced by site-directed mutagenesis (SDM) to accurately identify single-allele insertions (Fig. 1: step 3). To introduce the K700E mutation in exon 15 of the human *SF3B1* genomic locus, we introduced a G mutation (for AAA to GAA codon change) through SDM in the right homology arm (RHA) (Fig. 1: step 4). Cloning of the arms into rAAV acceptor and the SDM reaction to introduce the screening primer and desired K700E mutation were extremely efficient and took ~10 days to perform.

An important consideration must be made in designing the homology arms flanking the Neo[R] gene-trap cassette: if the target intron is too short, the cassette may be integrated too close to the 3' splicing acceptor site at the intron/exon junction, potentially disrupting 3' branchpoint recognition leading to a failure of splicing out of the gene-trap-containing intron upon Cre-activation. To avoid this possibility, the longer of the two introns flanking the exon of interest (in which to introduce the desired modification) should be preferred for introduction of the gene-trap cassette. In this case, intron 13 was chosen over intron 14 due to its larger size.

Two recipient cell lines were chosen, both harboring wild-type *SF3B1* (transformed human K562 erythroleukemia cells and doxycycline-immortalized human umbilical cord-derived erythroid progenitor-2 (HUDEP-2) cells). K562 cells were transfected with the rAAV virus and selected for G-418 resistance at 48 h for 10–12 days at which point resistant cells (evident as doublets) had emerged. Single-cell clones were selected with limiting dilution and crude DNA preparation (purified from cultures using standard methods) was analyzed by polymerase chain reaction (PCR) using the "In–Out" screening strategy outlined in the "Methods" section (Fig. 2a, b). In the *SF3B1* example, using the first screening primer located in the targeted genomic locus outside the RHA and the second primer located in the middle of the left homology arm (LHA), this PCR screening strategy would generate 1.5 Kbp in the case of unedited wild type, 1.5 Kbp, and 1.1 bp with monoallelic HR and 1.1 bp with biallelic HR. This screening strategy did not fail in our experience and we

found it to be a cheap and reliable method that demands very low cell numbers.

Of the 25 clones analyzed, only four were found to have the correct targeted integration (Fig. 3a). Furthermore, Sanger sequencing of their genomic DNA revealed that none of the four clones carried the A → G mutation (Fig. 3c). This suggested that, while the rAAV vector was utilized as a template for HDR, the mutation was not incorporated likely due to the large distance (~300 bp) between the obligatory Neo[R] cassette and the mutation (Fig. 3e). We concluded that using rAAV alone as a strategy to introduce single-allele point mutations is limited by a long distance between the selection cassette and the mutation site, since the homologous recombination likely occurs proximal to and terminates before reaching the mutated site on the AAV template.

**Combining CRISPR/Cas9 with rAAV vastly improves efficiency of introducing single-allele mutations at the correct genomic locus.** We postulated that lack of incorporation of the desired mutation, despite successful editing at the correct locus, could be overcome by directing the HDR to this site through a DSB in its immediate vicinity. We therefore designed single-guide RNA (sgRNA) close to the K700 locus and generated Cas9/sgRNA ribonuclear proteins (as described in "Methods"). After confirming efficiency of the sgRNA to introduce DSBs through T7EI cleavage assay[35] (Fig. S1), RNPs were transfected into K562 cells (and HUDEP2 cells by nucleofection) followed by transduction of rAAV 1 h later. Cells were then allowed to recover and selected for Neo resistance followed by single-cell cloning (as described for rAAV-only methodology; Fig. S2). Single-cell clones were then screened by PCR for single-allele insertion at the correct locus (Fig. 3b). Twenty-three of the 25 clones were noted to have correct insertion (targeted integration efficiency of 92%). Furthermore, all these clones were noted to incorporate the desired A → G mutation (Fig. 3d–e). The DSB introduced by Cas9 facilitates locus-specific HR of the nascent DNA strand with the rAAV donor (Fig. 3f). It has been previously reported that editing is most efficient when targeted site is within 15 bp (ideally <10) of the Cas9-induced DSB, with efficiencies rapidly dropping beyond 20 nucleotides from the DSB site[36].

Clones thus identified were expanded, transfected with Cre-recombinase plasmid, and single clones were selected (based on correct Cre-mediated recombination as per PCR strategy, Fig. 2c). RNA from these "post-Cre recombination" cells were analyzed for K700E mutation (through sequencing of cDNA amplicon), which showed comparable expression of wild-type and mutant transcripts (Fig. 4a). As shown in western blot (Fig. 4b), heterozygous SF3B1 cells, post-induction, had double the SF3B1 protein expression as compared to uninduced K562 isogenic cells due to expression of the mutant allele along with the unmodified allele. We further confirmed the functional expression of the mutant allele by analyzing the cDNA for the presence of aberrant 3' splice sites (cryptic 3'SS), the characteristic splicing aberration brought about by the *SF3B1* mutation[37,38]. *SF3B1* mutations result in widespread use of cryptic 3'SS a few base pairs upstream of the canonical sites in previously reported genes such as *UBR4*, *ABCB7*, *SEPT6*, *ENOSF1*, and *DYNLL1*; Fig. 4c).

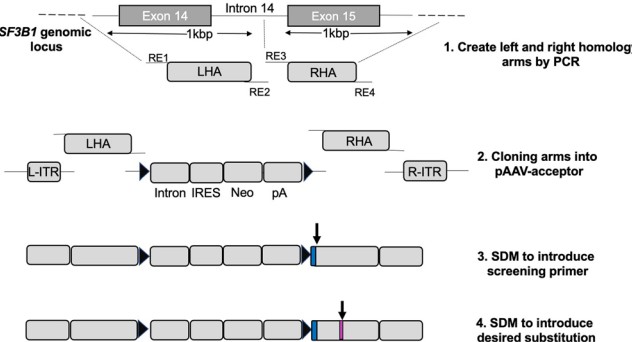

**Fig. 1 Schema for creation of the rAAV-donor.** Step 1: left (LHA) and right (RHA) homology arms were created by PCR from human genomic DNA template using PCR primers tailed with restriction enzyme sites (REs; RE 1, 2 for LHA and RE 3, 4 for RHA) for subsequently cloning into the pAAV acceptor. Step 2: The left and right homologous arms were digested and sequentially cloned into the pAAV-SEPT-Acceptor plasmid, which has been built with a Neomycin (Neo[R]) gene-trap cassette. Step 3: The extra 20 bp from upstream exon (primer 2 sequence) was introduced in the beginning of the RHA of the pAAV-SEPT-Acceptor by site-directed mutagenesis (SDM). Step 4: The desired substitution (K700E mutation) was introduced in the right arm of the pAAV-SEPT-Acceptor by SDM (shown in magenta).

**Inducible expression of Cre-recombinase overcomes toxicity of mutant SF3B1.** While we were able to reliably generate single-cell mutant clones using rAAV/CRISPR editing, we were not able to maintain them in culture beyond 2–3 weeks. Cells slowed in their growth, increased in size, and eventually became apoptotic. Our results were similar to data from previous studies which have noted toxicity of mutant splicing factors that resulted in silencing

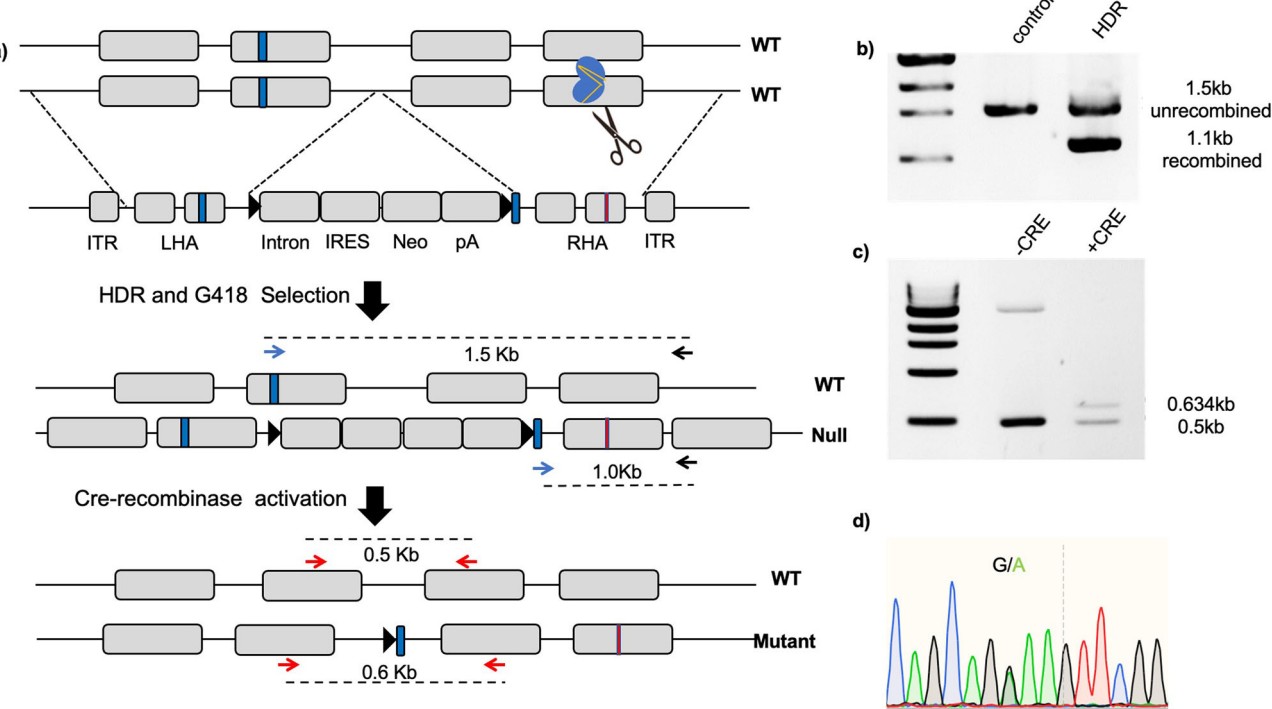

**Fig. 2 Schematic overview of single-allele editing of *SF3B1* locus using CRISPR/Cas9 and rAAV. a** Schematic of "In–Out" PCR screening strategy of clones after single-allele recombination before Cre activation (screening primer pair denoted as black and blue arrows) and after Cre activation (denoted as red arrows) is shown. **b** PCR analysis of genomic DNA isolated from unedited and edited K562 cells that underwent recombination (using screening primer pair from the middle panel of **a**). **c** PCR analysis of genomic DNA from edited K562 cells post Cre activation (using screening primer pair from the lower panel of **a**). **d** Confirmation of successful single-allele K700E mutation by Sanger sequencing of genomic DNA.

of transgene, loss of mutant allele, or cell death[9]. This is clearly a major obstacle in biochemical experiments that require large number of cells. We posited that, if mutant factors are expressed only after the clone is expanded to a sufficiently large number, this limitation could be overcome. Therefore, we opted to transduce the PCR-positive clones with inducible FLAG-tagged Cre-recombinase expressing lentiviral vectors (as described in the "Methods" section) to generate stable inducible Cre-recombinase-based isogenic cells. We found this strategy to successfully generate stably integrated, inducible Cre-recombinase cell lines. Cre-recombinase expression in the clones, upon doxycycline administration, was confirmed by western blot using anti-FLAG antibody (Fig. S3). Regulated induction of the Cre-expressing clones to activate Cre expression would remove the floxed Neo[R] cassette and allow expression of the targeted mutant allele (Fig. 2a, c, d). Upon doxycycline induction, G → A substitution was observed beginning at 2 days and noted to be equal to A allele by about day 5 (Fig. S4). We speculate that this is because, although the LoxP cassette is floxed promptly upon Cre-recombinase expression, the epigenetic memory of silencing marks on the "null" allele is not reversed until at least a round of cellular replication.

These inducible isogenic cells are currently being used for biochemical studies that investigate mechanisms of *SF3B1*K700E. We have used the strategy to generate similar *SF3B1*-mutant cells in a second hematopoietic cell line (HUDEP-2) and to generate isogenic cell lines involving oncogenic mutation of another splicing factor (S34F mutation in *U2AF1*) (Fig. S5). To further demonstrate the generality of our approach, we have also generated stable HeLa adherent cell lines with endogenous FLAG-epitope-tagged *SF3B1* gene (depicted in Fig. 5). Unlike in the case with mutations in *SF3B1*, pure FLAG-tagged-*SF3B1* isogenic cell lines can be generated and expanded after transient Cre expression without requiring an inducible Cre-recombinase system.

## Discussion

Isogenic model systems are critical to defining complex molecular mechanisms of clonal diseases such as cancer, where mutations are often limited to one allele. Overexpression models where mutated genes with neomorphic functions are expressed several folds of their wild-type counterparts are clearly of limited utility. In some instances, like oncogenic mutations of splicing factors, paradoxical growth suppression or toxicity is noted in vitro. These scenarios have created roadblocks in the generation of physiological models and need specialized genome-editing approaches capable of creating scalable isogenic cell lines at high efficiency.

Herein we establish a novel combinatory approach of CRISPR/Cas9 with rAAV gene targeting and demonstrate its utility in creating isogenic human cell lines.

Single strand DNA templates are associated with improved knock-in efficiency and fewer off-target insertions compared to double-stranded DNA donors[39–41]. Since our initial attempts to generate isogenic cell lines using conventional CRISPR/Cas9 and short ssDNA templates (~200 bp synthesized oligonucleotides) were unsuccessful, we did not systematically compare knock-in efficiencies of this approach to the one we describe here (i.e., rAAV with CRISPR/Cas9). Given the toxicity of mutant SFs, it is likely that, even when generated through extensive screening, these cell lines will likely lose the mutant allele or die out upon extended culture.

With regard to linear ssDNA oligomers versus rAAV donor templates, it must be noted that the rAAV templates have extremely high knock-in efficiencies due to the virus' natural ability to stimulate HR. As ssDNA viruses, rAAVs are naturally suited to be HDR templates. Additionally, the pAAV-SEPT-Acceptor plasmid uses a promoterless splice acceptor-IRES-Neo[R] gene cassette to select correctly integrated clones, which further enhances the HR efficiency. Combining CRISPR/Cas9 and rAAV

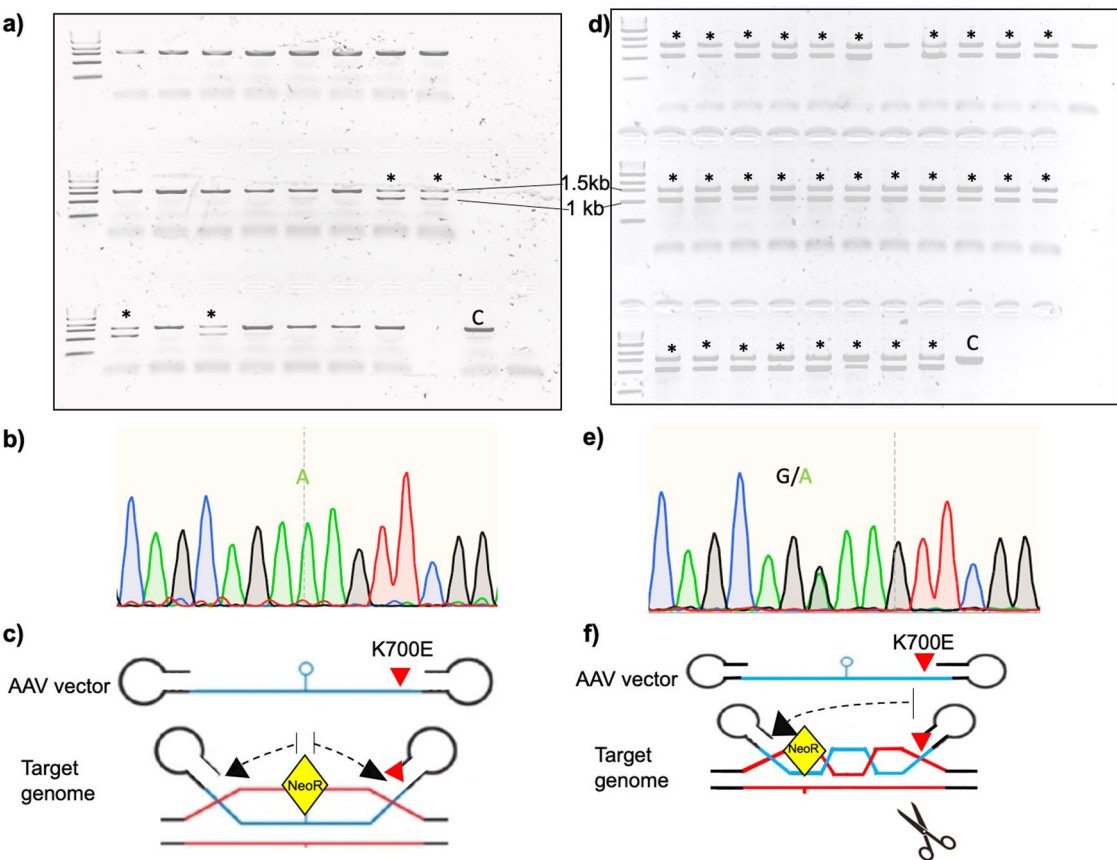

**Fig. 3 On-target integration events in G418-resistant clones by rAAV and rAAV-CRISPR/Cas9. a** PCR-based identification of on-target events in *SF3B1* alleles using rAAV-based homologous recombination alone strategy. Four of the 25 (16%) G418-resistant clones were PCR positive for on-target integration of the expression cassette (marked with asterisks); C control wild type. **b** Depicted is the Sanger sequencing of the gDNA target sequence of one of the representative PCR-positive clones generated by rAAV-alone-based HR. The clone did not undergo the desired K700E modification indicating failure of HR between the target DNA and rAAV in region of the desired modification (~300 bp from the gene-trap cassette integration). **c** Illustration of the HR event between rAAV vector and genome locus. In the absence of a targeted DSB, HR begins in the intron at the site of obligatory $Neo^R$ intron-trap insertion (depicted in yellow) spreading out with decreasing efficiency before falling off due to significant homology between the rAAV HAs and the host genome. **d** PCR-based identification of on-target events in *SF3B1* alleles using rAAV + CRISPR/Cas9-based homologous recombination strategy. Thirty-one of the 33 (94%) G418-resistant clones demonstrated on-target integration of the expression cassette (marked with asterisks). **e** Depicted is the Sanger sequencing of the gDNA target sequence of one of the representative PCR-positive clones generated by rAAV-CRISPR/Cas9-based HR, showing successful single-allele editing of the SF3B1 K700 locus. **f** The double-stranded break introduced by Cas9 facilitates locus-specific HR of the nascent DNA strand with the rAAV donor template containing the desired mutation.

in this fashion, we obtained >90% clones to be directed to the accurate genomic locus as compared to ~15% with rAAV alone, representing a significant improvement compared to other reports of rAAV-based gene editing[18,20,25]. To our knowledge, this is the first demonstration where CRISPR was used to specifically target the mutation site and a single rAAV donor served as the HDR template for both the desired modification as well as the gene trap. Our approach differs from the other CRISPR-AAV systems as it relies on HDR-mediated transgene cassette insertion in the adjacent intron but not at the Cas9 targeted DSB site.

While we attained recombination efficiencies of about 15% in the case of SF3B1 editing, it is notable that none of these clones incorporated the K700E mutation. We speculate that this is likely due to the distance of ~300 bp between the mutation site and intron-trap cassette. Without a DSB to direct the HDR, the HR process likely terminates proximal to the K700 locus (Fig. 3c, f). Introducing a targeted DSB with CRISPR/Cas9 avoids this problem by directing HDR to the vicinity of the K700E locus. Hence, in addition to achieving >90% homologous recombination efficiency, all the clones screened were positive for the G → A at the K700E locus.

While combining rAAV and CRISPR/Cas9 resulted in high efficiency of gene editing, toxicity of SF3B1-K700E resulted in death of these clones during their expansion. We overcame this by expressing Cre-recombinase from an doxycycline-inducible system. Thus, cells can be expanded prior to Cre recombination and induced to express Cre-recombinase. Editing efficiency upon doxycycline induction is nearly 100% after a few days, confirming utility of the strategy for biochemistry studies that require several million cells. To demonstrate the generalizability of our approach, we have also optimized the strategy in HUDEP-2, a non-transformed cell line that is difficult to edit compared to transformed cell lines such as K562.

While the intron-trap design can be adapted for most genes, the following considerations must be kept in mind. First, a robust PCR-based screening strategy must be in place before the editing of a particular gene. This may be problematic for GC-rich or low-complexity genomic sequences and may require optimization of primer design or cycling conditions. Second, the *cis* regulators of splicing (such as branch point, 3' and 5' splice sites) may be perturbed by transgene cassette insertion, especially when introns are small. Therefore, the boundaries of LHA and RHA must be

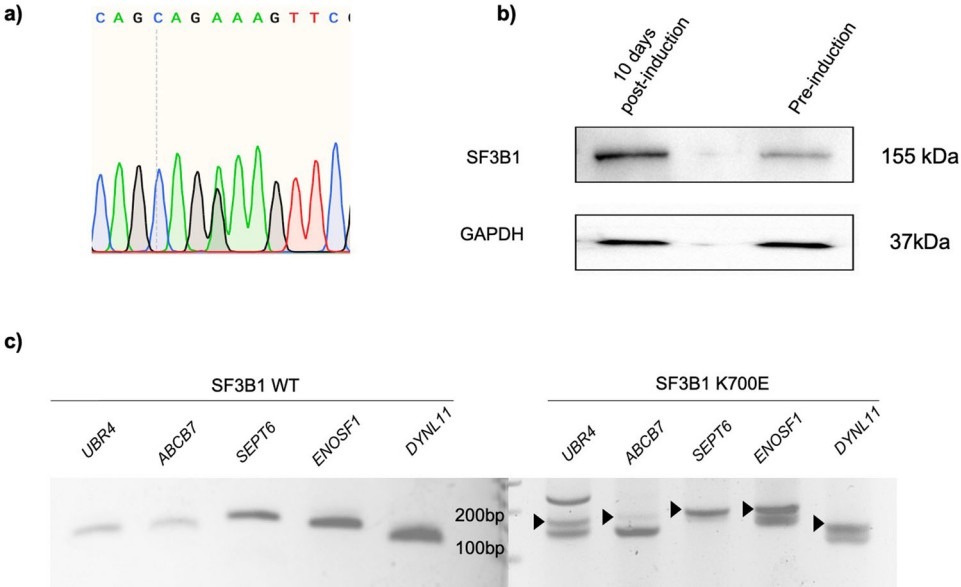

**Fig. 4 Validation of human cell lines with single-allele *SF3B1* mutant gene. a** Sanger sequencing of cDNA from K562 isogenic cell line sample (5 days post-induction) confirming mutant SF3B1 mRNA expression. **b** Western blot-based confirmation of expression of the mutant *SF3B1* allele. Depicted is the western blot with SF3B1 antibodies on lysates from parental (*SF3B1*wt/null) clone and cells 10 days post-induction (*SF3B1*wt/K700E). The induction of mutant *SF3B1* allele leading to expression of mutant SF3B1 protein (same size as the wild-type SF3B1 protein) doubles the SF3B1 protein expression as compared to the pre-induced sample. **c** Validation of cryptic 3′ splice site usage in five selected genes frequently mis-spliced in *SF3B1* mutant myelodysplastic syndrome. PCR products of the five genes (*UBR4, ABCB7, SEPT6, ENOSF1, DYNL11*) showing aberrant cryptic 3′SS usage in the K562 *SF3B1* mutant (cryptic transcripts indicated by black arrow heads).

carefully defined such that these sequences are not disturbed resulting in splicing failure. Third, the approach requires G418 selection of infected clones, making it unsuitable for in vivo applications and primary cell line editing. Fourth, unlike CRISPR–Cas9 approaches which can be multiplexed, the combinatory approach is not ideal for applications requiring simultaneous modifications of multiple genes in a cell or multiple loci within a single gene. Finally, it is important to keep in mind that transgene insertions may not only physically disrupt genes at their site of insertion[42] and influence gene function but also potentially have effects on function of neighboring genes[43,44]. Therefore, it is recommended to generate multiple cell lines of the same transgene and compare them for any unexpected phenotypes that may be due to the insertion site. It must be mentioned here that the strategy of using a drug resistance cassette flanked by LoxP sites within introns mitigates effects on the expression of neighboring genes by allowing for the removal of the NeoR minigene upon Cre-recombinase activation.

In addition to its utility to generate inducible single-allele mutants, the rAAV intron-trap CRISPR/Cas9 approach lends itself to other applications, such as creation of heterozygous endogenous epitope tags and single-allele knockouts. In summary, we demonstrate the feasibility combining rAAV and CRISPR for targeted gene editing that significantly improves upon conventional approaches for generation of isogenic cell lines, both in terms of precision and efficiency. Using an inducible Cre-recombinase also overcomes the issue of toxicity of the resulting mutant protein.

## Methods

**rAAV donor plasmid production**. All PCR primers used are listed in Table S1. To efficiently screen accurately recombined cells, a robust screening strategy is essential. An "In–Out" PCR screening strategy was employed as we have previously reported[18] (Fig. 2a). Briefly, the first screening primer is located in the targeted genomic locus outside one of the homology arms (RHA in this case) and the second primer is located within the other arm to amplify an ~1.5 Kbp fragment. A

copy of the second primer is inserted into the 5′ end of the RHA (adjacent to the NeoR trap cassette).

Homology arms for the human *SF3B1* gene were generated by PCR using human genomic DNA as template. LHA and RHA were approximately 1 Kbp in size. As illustrated in Fig. 1, LHA and RHA spanned a genomic region from chr2:197401496-197403593. LHA and RHA were cloned to the AgeI/SacI and ClaI/SalI sites of the pAAV-SEPT-Acceptor plasmid at the multiple cloning sites I and II, respectively. SDM for the insertion of the screening primer and generation of a point mutation in the RHA of the SF3B1 vector was sequentially performed using the Q5 SDM Kit (NEB Biolabs) with primers designed to create the screening primer and K700 locus (K700K → K700E) nucleotide substitution, respectively. The integrity of the plasmid was confirmed by restriction analysis and sequence verified by Sanger sequencing.

**rAAV virus production**. The rAAV vectors were produced using the AAV-Helper-free System protocol (Agilent Technologies) with modifications, as previously described[18]. Briefly, AAV-293 cells (HEK-293 cells transformed by adenovirus type 5 DNA)[45] were plated at $3 \times 10^6$ cells per 100-mm tissue culture plate in 10 ml of Dulbecco's Modified Eagle Medium (DMEM) growth medium (Gibco) supplemented with 10% (v/v) fetal bovine serum (FBS) (Thermofisher) and 1% penicillin–streptomycin (Gibco) in a humidified 5% $CO_2$ at 37 °C. At 48 h after seeding, the cells were co-transfected with 10 μg each of pAAV-SEPT, pAAV-RC (rAAV2 serotype; Stratagene), and pAAV-Helper (Stratagene) plasmids using a standard calcium phosphate-based protocol, as previously described[46]. At 72 h post transfections, cells were harvested and the cell suspensions in the DMEM growth medium were subjected to lysis through four rounds of freeze/thaw by alternating the tubes between a dry ice-ethanol bath and the 37 °C water bath. The lysates were centrifuged at $10,000 \times g$ for 10 min at room temperature (RT). The lysis supernatant was transferred to a fresh tube and viral stocks were stored at −80 °C.

Titration of the rAAV-2 viral stock was carried out as previously published[47]. Briefly, to determine the viral genomic (VG) physical titer per ml, we treated 10 μl of purified virus with DNAse1 (NEB M0303S) following purification of viral DNA using the DNAeasy Blood and Tissue Kit (Qiagen 69506) as per the manufacturer's protocol. A standard curve was obtained using the pAAV-SEPT purified plasmid by using serial 10× dilutions of $10^8$–$10^4$ copies. Quantitative PCR (qPCR) was performed on BioRad CFX96 using KAPA qPCR mix. The DNAseI-treated VG titers were determined to be $6 \times 10^7$ copies/ml and $7.3 \times 10^8$ copies/ml for rAAV-*SF3B1*K700E and rAAV-*U2AF1*S34F (discussed later) viral supernatants, respectively (see Table S3). Estimation of rAAV VG titers is generally recommended to ensure reproducibility of transduction efficiency. Given the relatively high transduction rates associated with rAAV2[48], we chose to infect all our study cell lines (K562, HUDEP-2, HeLa) with a constant multiplicity of infection (MOI) of ~500.

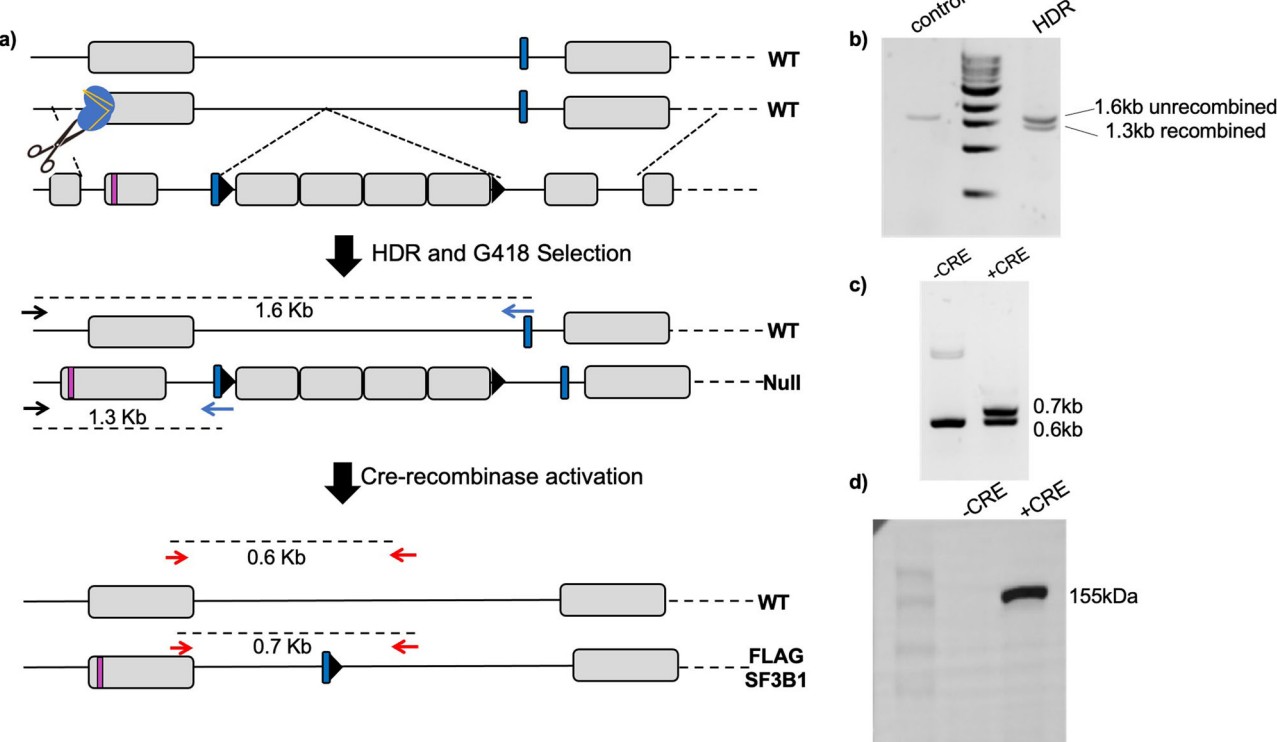

**Fig. 5 Single-allele epitope tagging of SF3B1 using CRISPR/AAV in HeLa cells. a** Schematic of genomic *SF3B1* locus and rAAV vector is shown. AAV vector consists of left homology arm (LHA) with FLAG-tag and extra 20 bp from downstream exons added to the end for PCR screening, neomycin cassette with IRES, poly(A) termination signal, and a right homology arm (RHA). LoxP sites (shown by black triangles) were placed as indicated to excise out the neomycin cassette after recombination was confirmed. Initially a null allele is formed due to termination of transcript at the p(A) signal following the NeoR cassette. Cre recombination leads to elimination of the cassette flanked by two LoxP sequences and the FLAG-tagged allele is expressed again, with a FLAG tag at N-terminus. **b** PCR analysis of genomic DNA isolated from unedited and edited K562 cells that underwent recombination. **c** PCR analysis of genomic DNA from edited K562 cells pre- and post-Cre activation. **d** Confirmation of successful FLAG-tagged SF3B1 by western blot with FLAG antibody.

**Cas9 protein expression**. The Cas9 nuclease from *Streptococcus pyogenes* (Spy-Cas9) was expressed in *Escherichia coli* (NEB #M0386) and purified[49]. The protein was stored at −20 °C in a preservation buffer containing 10 mM Tris-HCl, 300 mM NaCl, 1 mM DTT, 0.1 mM EDTA, and 50% glycerol.

**sgRNA in vitro transcription**. The sgRNAs were designed using the GPP sgRNA Designer online webtool (https://portals.broadinstitute.org/gpp/public/analysis-tools/sgrna-design). The DNA template encoding the sgRNA containing the targeting sequence in the region of interest was assembled using synthetic oligonucleotides, as previously described[50]. The sgRNAs were generated by in vitro transcription of DNA template using the HiScribe T7 Quick High-Yield RNA Synthesis Kit (NEB #E2050). The sgRNA sequences are provided in Table S2.

**Cell culture**. Human K562 erythroleukemia cells and HUDEP-2 cells were used for *SF3B1* editing. The K562 cells were established from human chronic myelogenous leukemia cells as described previously[51], tested free of mycoplasma contamination, and short-tandem repeat validated by ATCC. Cells were maintained in a medium containing 90% RPMI 1640 (Gibco) supplemented with 10% FBS (Thermofisher) and 1% penicillin–streptomycin (Gibco). The HUDEP-2 cells were established from human umbilical blood as described previously[52] and provided as a kind gift from Dr. Nakamura[53]. Cells were maintained in a HUDEP-2 expansion medium[54].

**Cas9 RNP and rAAV delivery**. K562 cells were seeded in 12-well plates at a density of $2.5 \times 10^5$ cells per well. After 24 h, 6 nM of Cas9 nuclease (0.3 μl of 20 μM stock) was mixed with 12 nM of guide RNA (0.3 μl of 40 μM stock), in a 1:2 molar ratio, diluted with 100 μl of Opti-MEM Reduced-Serum Medium (Gibco), followed by addition of TransIT-X2® (2 μl) to generate TransIT-X2®:RNP complexes (Mirus Bio). The TransIT-X2®:RNP complex mixture was then added to the cells. After 1h incubation at 37 °C, cells were spinfected with the rAAV viral supernatant (volume determined by MOI) at 1800 RPM for 45 min at RT. Cells were maintained in serum-containing medium in a humidified 5% $CO_2$ at 37 °C.

The HUDEP-2 cells were seeded at a density of $2.5 \times 10^5$ cells, per well. After 24 h, cells were harvested, centrifuged at $300 \times g$ for 5 min, washed once with phosphate-buffered saline, and resuspended in 20 μl of electroporation Solution

(Mirus Ingenio) in a 0.2 cm cuvette. In a separate tube, 4 μl of Cas9 nuclease was mixed with 5 μl of guide RNA in a 1:2 molar ratio (1500 nM gRNA:750 nM Cas9; final concentration in cuvette) and incubated for 5 min at RT. The RNP complex mixture was subsequently added to the cell mixture and cells were nucleofected using the Amaxa® 96-well Shuttle system (Lonza) and the program U-008. Immediately after nucleofection, 200 μl of serum-containing medium was added to the nucleofector cuvettes, and cells were transferred onto 24-well plates and incubated in HUDEP-2 expansion medium at 37 °C. After 1-h incubation at 37 °C, cells were spinfected with the rAAV vector (volume determined by MOI) at 1800 RPM for 45 min at RT. Cells were maintained in serum-containing medium in a humidified 5% $CO_2$ at 37 °C.

**Identification of the mutant clones**. Two days after spinfection, G418 (Sigma) at a final concentration of 0.2 mg/ml was added to the cell culture medium. The cell culture media with G418 was replenished every 3 days. Addition of antibiotics was terminated after 2 weeks at which time resistant clones emerged. To isolate the live cells, Ficoll gradient separation with Histopaque-1077® (Sigma) was used. Single-cell clones were then isolated using limited dilution into 96-well plates in the presence of G418 at 0.2 mg/ml. Individual G418R clones were expanded and then checked for homologous recombination of the targeting rAAV vector using PCR primer pairs specific for the targeted allele using a PCR screening strategy described in the "rAAV donor plasmid production" section.

**Generation of inducible SpCas9/CRISPR cell line systems**. To generate the inducible Cre-recombinase-based cell lines, K562 (inducible for Cre with doxycycline) and HUDEP-2 cells (inducible for Cre with tamoxifen [doxycycline is already a component of the HUDEP-2 expansion medium]) were transduced with lenti-viral vectors encoding Cre-2A-Puro (Tet-pLKO-Puro [Addgene # 21915] for K562; MSCV-CreERT2-Puro [Addgene # 22776] for HUDEP-2) followed by selection in a serum-containing media with 1 μg/ml of puromycin (Sigma) and 0.2 mg/ml of G418. Individual clones were expanded by limiting dilution and the Cre-recombined clones were identified through PCR of genomic DNA using PCR primer pairs specific for the targeted allele. Genomic editing was confirmed at the

K700 locus through Sanger sequencing of PCR amplicons, which demonstrated a mixed pattern of AAA (lysine) and GAA (glutamic acid).

Finally, a fraction of the cell population from each of the clones were washed, resuspended in culture medium containing puromycin 1 µg/ml but without G418 (since G418 sensitivity is restored with floxing of G418-resistance LoxP cassette upon Cre-recombinase induction), and induced with 1 µg/ml doxycycline (Sigma) or 1 mM 4-OH-tamoxifen (Cayman chemicals) in ethanol for K562 and HUDEP2, respectively. Confirmation of mRNA expression of the mutant *SF3B1* allele was made based on Sanger sequencing of cDNA prepared from RNA extracted from cell lysates, 2–7 days after induction. RNA was extracted using the RNeasy Mini Kit (Qiagen). cDNA synthesis was performed using standard PCR-based techniques. The primer sequences used in this and the previous section are provided in Table S1. For a more easy-to-follow protocol version format, the reader may refer to https://www.protocols.io/edit/high-efficiency-gene-editing-using-adeno-associate-bv99n996.

**Reporting summary**. Further information on research design is available in the Nature Research Reporting Summary linked to this article.

## Data availability

The datasets generated and/or analyzed during the current study not included in this published article (and its supplementary information files) are available from the corresponding author on reasonable request.

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

## Acknowledgements

We would like to thank the Yale Center for Genome Analysis (YCGA) and Yale Keck Center for DNA sequencing. The work was supported in part by National Institutes of Health (NIH) grants HL133406 (to M.M.P. and K.M.N.) and T32 CA233414 (to P.C.B.), a pilot grant from the NIDDK Core Centers for Excellence in Hematology (to P.C.B.)., a U54 pilot grant from YCCEH (A.K.G). and a pilot grant from the Frederick A DeLuca Foundation (to M.M.P.).

## Author contributions

P.C.B and A.K.G. designed and performed experiments and wrote the manuscript. J.-S.K., T.W., and K.M.N. provided experimental guidance and edited the manuscript. M.M.P. was involved in designing the strategy and writing and editing the manuscript.

## Competing interests

The authors declare no competing interests.
