## [Transparent Peer Review File · Communications Biology]

Reviewers' comments:

Reviewer #1 (Remarks to the Author):

The authors describe the development of a CRE inducible editing system designed to be used in cell lines. The system is designed to integrate foreign DNA into a desired locus which creates a null allele. However, it also is designed to integrate a desired mutation. Once Cre is added to the system, a portion of the foreign DNA is excised and the allele is able to be expressed correctly, but this time as the modified version. It's a clever system to overcome the barriers of attempting to create potentially toxic mutations that would limit clonal expansion.

The experiments for the most part are solid and technically sound. However, I have made several comments that can be easily address with some editing of the manuscript.

My comments are below and are not listed in any order of importance.

What isn't clear from the current experiments, is how efficient is HDR with just CRISPR/Cas and a linear donor template versus CRISPR/Cas and AAV delivering the donor template. Considering that the system utilized requires G418 selection, I would assume that efficiencies of obtaining correctly edited cells would be high regardless. This isn't addressed in the manuscript.

The writing in the manuscript could be improved for readability. While most of it is fine, there are sentences that appear clumsy.

Some Examples:

Such scenarios include generation of single-allele missense point-mutations and/or when the resultant mutation confers a physiological growth disadvantage. And/or?

It is therefore critical to study these mutations in isogenic model systems with one allele mutation, and not in over-expression models where the proportion of wild-type to mutant protein is unlikely to be equal. One allele mutation?

The authors state on line 70, " the AAV vector is transcribed to a long single stranded DNA template that serves as an efficient template for HDR following double stranded DNA breaks introduced by CRISPR/Cas9". To me this reads as an incorrect statement. Transcription is not the correct term here. Additionally, please clarify that it is the double stranded AAV genome that serves as the template for HDR rather than the single stranded AAV genome, or vice versa or both.

The authors appear to write about AAV from the perspective that it integrates into the genome efficiently. It's true that some have used AAV to integrate DNA into the genome. But I think the introduction would benefit with paragraph placing into context the use of AAV as a method of integrating foreign DNA into the genome. For example, WT AAV that carries the Rep genes can integrate into the genome. But the AAVs used in labs that typically do not carry the Rep genes do not integrate into the genome efficiently. They exist mainly in cells as extra-chromosomal episomes. But they can integrate into the genome randomly at very low frequency. It is my understanding that AAV can be used to deliver the HDR template. But in this case the AAV is used as a delivery vehicle for the DNA template - nothing else. The authors should revise the introduction to make this clear. Or discuss the evidence to the contrary.

The systems the authors are using rely on G418 selection? What % of integration do you see without selection? This should be discussed in the manuscript.

The authors state, "Despite high efficiency of AAV-HDR, by up to more than 10- fold over conventional plasmid-based expression systems 22 ." It would be helpful for the reader to state the actual

efficiencies detected so there is context for what "10-fold" really means. Also, what is considered "high-efficiency". Is it the fact that the AAV is delivering the template in a linear form vs a plasmid that is circular? Some more details here could be helpful.

The authors state on line 78, "adaptations of the AAV-CRISPR gene-editing methodology with varied in vivo and in vitro applications have since been reported 20,21,23-26 , all of which rely on targeted transgene insertions. This statement is NOT correct. Some of these for sure are papers that do not report on systems that require targeted insertions. Once again AAV mediated CRISPR systems do not require any integration to mediate Indel or HDR formation.

It doesn't appear that the AAVs were titered. This could create problems for reproducibility of the findings. The authors should titer their viruses.

It does not appear that there was a DNase treatment of the viral supernatant. Therefore, how do we know the reported results are actually due to functional AAVs and not just free AAV genomes or plasmid DNA in the supernatant? Also are you getting more AAV viral genome integration due to pRC plasmid contamination?

The serotype of the AAV pRC plasmid is not stated. It should be.

To increase readability, it would be beneficial to write about figure 1 in the results section too.

I would recommend to explain how the "AAV model with intron trap" functions regardless if you have previously described it elsewhere in the interest of readability.

Reviewer #2 (Remarks to the Author):

In their paper the authors combine AAV, CRISPR and inducible Cre recombinase to introduce oncogenic mutations into cell lines in culture showing that this system has enhanced efficiency when compared to other systems or CRISPR/Cas9 alone. I have several specific comments on this paper as detailed below.

1. The approach developed is interesting and thoughtful however the authors only provide data for the use of their method at the SF3B1 locus. The obvious question is how scalable/useful is this approach at multiple loci or randomly selected loci – there is mention of an experiment involving editing of U2AF1 at S34F but the data and success of that experiment is not provided). In the same way editing is attempted in K562 cells and also HUDEP-2 cells – both cell lines are relatively easy to manipulate with CRISPR so I am also left wondering how robust this method is across other cell types including hard to edit cell lines. Where there is a real opportunity for high-efficiency editing methods is in the generation of isogenic primary cell lines/cultures. Can the authors show editing in primary cells?
2. It is well known that inserting transgenes/markers at a locus can influence gene function (see <https://www.jimmunol.org/content/169/12/6875> for example). Although deletion of the LoxP sites results in the removal of these elements it might be useful to at least mention this caveat since it has important implications for which cells/alleles are used as controls in any experiment using isogenic lines generated with this method.
3. I presume all of the necessary plasmids have been deposited in Addgene? i.e. those generated as part of this paper.
4. Were the cell lines STR profiled/validated? Not clear from the text.
5. It would be good for the authors to provide all of the step by step protocols through Protocols .IO (or equivalent). As written it would be hard for a reader to perform all of the required steps.
6. There are no marker sizes on Figure 3a and 3d. Similarly on Figure IV b & c.

Dear editor and reviewers,

We thank you for the consideration of our manuscript and the insightful critiques to help improve the manuscript. We have attempted to answer to all the critiques to the best of our abilities or provided explanations as to why we are unable to address the comment experimentally. Below are detailed replies to each of the reviewers' comments.

Reviewer #1 (Remarks to the Author):

The authors describe the development of a CRE inducible editing system designed to be used in cell lines. The system is designed to integrate foreign DNA into a desired locus which creates a null allele. However, it also is designed to integrate a desired mutation. Once Cre is added to the system, a portion of the foreign DNA is excised and the allele is able to be expressed correctly, but this time as the modified version. It's a clever system to overcome the barriers of attempting to create potentially toxic mutations that would limit clonal expansion.

The experiments for the most part are solid and technically sound. However, I have made several comments that can be easily address with some editing of the manuscript.

My comments are below and are not listed in any order of importance.

1. What isn't clear from the current experiments, is how efficient is HDR with just CRISPR/Cas and a linear donor template versus CRISPR/Cas and AAV delivering the donor template. Considering that the system utilized requires G418 selection, I would assume that efficiencies of obtaining correctly edited cells would be high regardless. This isn't addressed in the manuscript.

It would be indeed useful to know the differences in efficiencies of the model we describe vs. a conventional one where in the CRISPR/Cas9 is combined with a linear donor template. We had tried to generate SF3B1-K700E with such a conventional CRISPR/Cas9 with linear template (ultramer from IDT). but were unable to successfully isolate edited clones despite screening several hundred clones. It is likely that this was from the toxicity of the mutated protein. We hence turned to AAV and later to AAV-CRISPR/Cas9 combination. Thus, the method was built on our inability to use CRISPR/Cas9 plus linear donor template. We have emphasized this in our revised manuscript (introduction section) (page 3, lines 98-102).

It is also correct that addition of the Neo cassette likely increases the efficiency from the ability to select resistant clones. In this context, AAV vectors have extremely high knock-in efficiencies due to the virus natural ability to stimulate HR by serving as single stranded DNA templates which is ideally suited for strand invasion in the HR pathway process. We have now rewritten the discussion with the above information in mind (page 13, lines 438-454).

The writing in the manuscript could be improved for readability. While most of it is fine, there are sentences that appear clumsy.

Some Examples:

2. Such scenarios include generation of single-allele missense point-mutations and/or when the resultant mutation confers a physiological growth disadvantage. And/or?

Thank you for the comment. We have rephrased the sentence

3. It is therefore critical to study these mutations in isogenic model systems with one allele mutation, and not in over-expression models where the proportion of wild-type to mutant protein is unlikely to be equal. One allele mutation?

Thank you for the comment. We have deleted the phrase-“one allele mutation”

4. The authors state on line 70, “ the AAV vector is transcribed to a long single stranded DNA template that serves as an efficient template for HDR following double stranded DNA breaks introduced by CRISPR/Cas9”. To me this reads as an incorrect statement. Transcription is not the correct term here. Additionally, please clarify that it is the double stranded AAV genome that serves as the template for HDR rather than the single stranded AAV genome, or vice versa or both.

Thank you for the correction. We agree that using the term ‘transcription’ is incorrect in this context. We have corrected our statement. It is the single stranded AAV genome which serves as the template. We have clarified this further in the introduction section

5. The authors appear to write about AAV from the perspective that it integrates into the genome efficiently. It’s true that some have used AAV to integrate DNA into the genome. But I think the introduction would benefit with paragraph placing into context the use of AAV as a method of integrating foreign DNA into the genome. For example, WT AAV that carries the Rep genes can integrate into the genome. But the AAVs used in labs that typically do not carry the Rep genes do not integrate into the genome efficiently. They exist mainly in cells as extra-chromosomal episomes. But they can integrate into the genome randomly at very low frequency. It is my understanding that AAV can be used to deliver the HDR template. But in this case the AAV is used as a delivery vehicle for the DNA template - nothing else. The authors should revise the introduction to make this clear. Or discuss the evidence to the contrary.

Thank you for pointing out the oversight; AAV genomes indeed typically are non-integrating. We have highlighted this in the rewritten introduction and discussion section (page 3, lines 114-118).

6. The systems the authors are using rely on G418 selection? What % of integration do you see without selection? This should be discussed in the manuscript.

Thank you for the comment. In the absence of the added advantage of an intron trap, AAV is like any other single stranded HDR donor. We anticipate that the percentage of integration using CRISPR + AAV, but without G418 selection, would be in the range expected with CRISPR/Cas9 and ssdonor template which, despite being better than a dsdonor template donor, is still low given the inefficiency of the HDR process. The advantage with using the pAAV-SEPT-Acceptor as the donor template is that it contains a promoterless splice acceptor-IRES-Neo^R gene cassette which significantly enhances the homologous integration efficiency by gene trap enrichment. It is precisely the G418 selection process which is significantly improving the homologous integration efficiency. We have not addressed this specifically in this manuscript since the AAV-intron trap system is built up on the previous manuscript from Waldman lab that described the advantages of such a promoter-less system to express Neomycin resistance cassette (Kim JS, NAR 2008). We have added additional text in the revised manuscript directing readers to the previous manuscript

7. The authors state, “Despite high efficiency of AAV-HDR, by up to more than 10- fold over conventional plasmid-based expression systems 22 .” It would be helpful for the reader to state the actual efficiencies

detected so there is context for what “10-fold” really means. Also, what is considered “high-efficiency”. Is it the fact that the AAV is delivering the template in a linear form vs a plasmid that is circular? Some more details here could be helpful.

The authors state on line 78, “adaptations of the AAV-CRISPR gene-editing methodology with varied in vivo and in vitro applications have since been reported 20,21,23-26 , all of which rely on targeted transgene insertions. This statement is NOT correct. Some of these for sure are papers that do not report on systems that require targeted insertions. Once again AAV mediated CRISPR systems do not require any integration to mediate Indel or HDR formation.

Thank you for the valuable comments. The 10-fold differences in ref #26 refers to the gene-editing efficiency of the RNP + ss linear AAV donor system as compared to RNP+ conventional double stranded plasmid donor system (this was based on percentage of cells expressing GFP-tagged construct derived from HR). We have added more details in the introduction section to make it clearer. Based on the reviewer’s suggestions, we have rephrased the statement on “adaptations...insertions”. We acknowledge that not all papers report on systems requiring targeted insertions. We have edited the introduction section accordingly (Page 4, lines 130-144).

8. It doesn’t appear that the AAVs were titered. This could create problems for reproducibility of the findings. The authors should titer their viruses.

Thank you for the valuable comment. We have titered the viruses. The data has been included under the methods section (“AAV virus production”) (Page 6/7, lines 215-225).

9. It does not appear that there was a DNase treatment of the viral supernatant. Therefore, how do we know the reported results are actually due to functional AAVs and not just free AAV genomes or plasmid DNA in the supernatant? Also are you getting more AAV viral genome integration due to pRC plasmid

Thank you for the observation. We acknowledge that plasmids directly taken up by cells serving as templates for HDR, and direct integration of AAV to the genome are theoretically possible. We think this is unlikely to contribute to functional HDR due to two reasons: First, K562 or HUDEP-2 cells are expected to take up only the encapsidated viral genomes and are not expected to take up free plasmids to any significant degree without additional manipulation. Second, random viral genome integration are very low frequency events with the rAAV2 delivery system. The single stranded AAV viral genome is far more likely to serve as a repair template than the double stranded AAV plasmids. The revised manuscript has clarified these points in the introduction and discussion sections in more detail.

10. The serotype of the AAV pRC plasmid is not stated. It should be.

Thank you for the observation. We have stated the serotype of AAV pRC in the methods section.

11. To increase readability, it would be beneficial to write about figure 1 in the results section too.

Thank you for the suggestion. We have written about figure 1 in the results section as well (page 10, lines 347-352).

12. I would recommend to explain how the “AAV model with intron trap” functions regardless if you have previously described it elsewhere in the interest of readability.

Thank you for the comment. We have discussed the specifics of the AAV-intron trap in the introduction section (page 3- lines 113-118).

Reviewer #2 (Remarks to the Author):

In their paper the authors combine AAV, CRISPR and inducible Cre recombinase to introduce oncogenic mutations into cell lines in culture showing that this system has enhanced efficiency when compared to other systems or CRISPR/Cas9 alone. I have several specific comments on this paper as detailed below.

1. The approach developed it interesting and thoughtful however the authors only provide data for the use of their method at the SF3B1 locus. The obvious question is how scalable/useful is this approach at multiple loci or randomly selected loci – there is mention of an experiment involving editing of U2AF1 at S34F but the data and success of that experiment is not provided).

Thank you for the comment. We would like to clarify that the method was optimized specifically to introduce mutations that are found in tumors (such as splicing factor mutations) that are toxic in cell line models. It is not intended as a high efficiency technique for multiplexed editing or to introduce mutations at random loci. To address generalizability, we have included a second locus (mutation in S34 locus of U2AF1, another splicing factor commonly mutated in cancer, and causes arrested growth in cell lines). We would also like to point out that multiplexing would be deleterious in the context of mutant splicing factors: mutating multiple factors simultaneously will be even more toxic to cells. We have clarified these points in the revised manuscript (page 15/16, lines 455-472).

2. In the same way editing is attempted in K562 cells and also HUDEP-2 cells – both cell lines are relatively easy to manipulate with CRISPR so I am also left wondering how robust this method is across other cell types including hard to edit cell lines. Where there is a real opportunity for high-efficiency editing methods is in the generation of isogenic primary cell lines/cultures. Can the authors show editing in primary cells?

Thank you for the comment. We would like to clarify that the technique is intended towards introduction of difficult to introduce isogenic mutation in cell lines , a problem that has limited biochemical studies in the field. Genome-wide profiling beyond RNA-seq are not feasible to be conducted in primary patient tissues or primary cells since several million cells are typically needed. Our technique also requires selection in G418 for several weeks, which is not feasible in primary cells. We have optimized the technique in HUDEP2, a non-transformed cell that is difficult to culture and hence manipulate; this was done to show broader generalizability of the approach. To further demonstrate generalizability, we introduced a FLAG-tag using AAV-intron trap-CRISPR which was done in HeLa to show that this approach can also be done in adherent cells. In the revised manuscript, we have highlighted that the approach is limited to cell lines and not for primary cells (page 14/15, lines 482-509).

3. It is well known that inserting transgenes/markers at a locus can influence gene function(see <https://www.jimmunol.org/content/169/12/6875> for example). Although deletion of the LoxP sites results in the removal of these elements it might be useful to at least mention this caveat since it has important implications for which cells/alleles are used as controls in any experiment using isogenic lines generated with this method.

Thank you for the comment. As per the reviewer's suggestions, we have highlighted these aspects in the discussion section (page 14/15,lines 482-509).

4. I presume all of the necessary plasmids have been deposited in Addgene? i.e. those generated as part of this paper.

Thank you for the suggestion. We are in the process of depositing the AAV-SEPT-SF3B1-K700E and AAV-SEPT-U2AF1-S34F to addgene

5. Were the cell lines STR profiled/validated? Not clear from the text.

Yes, the K562 cell lines have been STR validated by ATCC. HUDEP-2 cells do not have an ATCC designation and were provided to us as a kind gift from (Kurita, Nakamura et al PLoS One. 2013; 8(3): e59890)

6. It would be good for the authors to provide all of the step-by-step protocols through Protocols .IO (or equivalent). As written, it would be hard for a reader to perform all of the required steps.

Thank you for the suggestion. As per the reviewer's suggestion, we have drafted an easier to follow protocol.io version (page 9, lines 306-308). The version will be shared publicly pending finalization of the publication status of the manuscript.

7. There are no marker sizes on Figure 3a and 3d. Similarly on Figure IV b & c.

Thank you for the observation. We have added the marker sizes for figures 3 and 4

REVIEWERS' COMMENTS:

Reviewer #1 (Remarks to the Author):

No further comments